# Programmed Aptamer Screening, Characterization, and Rapid Detection for α-Conotoxin MI

**DOI:** 10.3390/toxins14100706

**Published:** 2022-10-14

**Authors:** Han Guo, Bowen Deng, Luming Zhao, Yun Gao, Xiaojuan Zhang, Chengfang Yang, Bin Zou, Han Chen, Mingjuan Sun, Lianghua Wang, Binghua Jiao

**Affiliations:** Department of Biochemistry and Molecular Biology, College of Basic Medical Sciences, Naval Medical University, Shanghai 200433, China

**Keywords:** alpha-conotoxin MI (CTX-MI), magnetic beads SELEX (MB-SELEX), high-throughput sequencing (HTS), aptamer, computer simulation; aptasensor

## Abstract

Conotoxins (CTXs) are a variety of mixed polypeptide toxins, among which α-conotoxin MI (CTX-MI) is the most toxic. Serious toxic symptoms, a lack of counteracting drugs, and cumbersome detection processes have made CTX-MI a hidden danger for humans. One of the obstacles to resolving this problem is the absence of specific recognition elements. Aptamers have shown great advantages in the fields of molecule detection, drug development, etc. In this study, we screened and characterized aptamers for CTX-MI through a programmed process. MBMI-01c, the isolated aptamer, showed great affinity, with an affinity constant (K_D_) of 0.524 μM, and it formed an antiparallel G-quadruplet (GQ) structure for the specific recognition of CTX-MI. Additionally, an aptasensor based on the biolayer interferometry (BLI) platform was developed and displayed high precision, specificity, and repeatability with a limit of detection (LOD) of 0.26 μM. This aptasensor provides a potential tool for the rapid detection of CTX-MI in 10 min. The aptamer can be further developed for the enrichment, detoxification, and biological studies of CTX-MI. Additionally, the programmed process is applicable to screening and characterizing aptamers for other CTXs.

## 1. Introduction

*Conus* species are carnivorous gastropods living in tropical and subtropical oceans [1]. Their venom ducts and poison glands secrete varieties of toxic polypeptides during predation and defense, which are called conotoxins (CTXs) [2]. Enormous gene families, abundant cysteines, and various post-translational modifications contribute to the diversity of CTXs [3,4]. According to statistics, the number of different sequences of CTXs has far exceeded 500,000 [3]. On the basis of signal peptides and disulfide bond skeletons, CTXs are divided into 26 superfamilies, which are represented by capital English letters; each superfamily is divided into several families based on the pharmacological targets, which are represented by lowercase Greek letters [2,5].

As a class of neurotoxins, CTXs act on a variety of ion channels and receptors throughout the nervous system, such as acetylcholine receptors and voltage-gated Na^+^ channels, with high selectivity and potency [6]. The sting of cone snails can cause paralysis, dizziness, convulsion, respiratory failure, or even death [2,7]. Dating back to the 1960s, more than 300 poisonings, including 30 deaths, have been reported [6,8,9]. Alpha-conotoxin MI (CTX-MI), which was first isolated from *Conus magus*, is considered as the most venomous species [10]. CTX-MI is a short peptide of 14 amino acids (GRCCHPACGKNYSC-NH_2_), which forms two disulfide bonds (Cys3-Cys8 and Cys1-Cys14) and selectively inhibits muscular acetylcholine receptors [11]. Due to a lack of clinical therapy and the unknown curative effect of antitoxic serums, the robust virulence of CTX-MI creates difficulties for prevention and clinical treatment [12]. In addition, the detection methods of CTX-MI rely on chromatography and mass spectrometry (MS) [13]. These methods are generally sensitive and precise. Nevertheless, the costly equipment, laborious sample purification, complex operation processes, and long testing time create barriers to their extensive applications [14]. The absence of specific recognition elements has been an obstacle to rapid detection and targeted therapy. As CTX-MI is a micro-molecular polypeptide with antigenicity but not immunogenicity, it is incapable of obtaining corresponding antibodies directly by immunizing animals. This deficiency of antibodies prompted us to screen novel recognition elements.

Aptamers, which are called “chemical antibodies”, are short single-strand DNA or RNA oligonucleotides with stable and unique three-dimensional (3D) structures [15]. For nearly one decade, multiple aptamers for small molecules have been successively screened [16,17]. These aptamers specifically tether targets with high affinity, suggesting that aptamers can be used as recognition elements for detection, detoxification, and biological research. Aptamers are usually screened by an in vitro approach called the systematic evolution of ligands by exponential enrichment (SELEX), which can exponentially enrich target-specific oligonucleotides from random libraries through rounds of screening and PCR amplification [18,19]. Aptamers have almost all the advantages of antibodies. More importantly, aptamers have unique features, including reproducibility, high thermal stability, low immunogenicity and toxicity, a lower cost of production, and permissible modification [16]. Hence, aptamers are expected to be applied in the detection, drug modification, and neurobiological research of CTX-MI.

In this study, we programmed screened, and characterized the specific and high-affinity aptamer for CTX-MI and verified its potential applications in rapid detection. The process was divided into four steps. (1) Candidate aptamers were enriched and isolated by magnetic beads SELEX (MB-SELEX). (2) Sequence diversity and the enrichment of candidate aptamers were analyzed using high-throughput sequencing (HTS) analysis. According to the HTS data, a total of 98 alternative sequences were screened. (3) The affinity and specificity of aptamers were determined by a biolayer interferometry (BLI) assay. Among these candidate aptamers, MBMI-01 had the highest affinity and specificity. Subsequently, the shorter aptamer, MBMI-01c, was obtained by removing the fixed regions. (4) The interaction mechanism was studied using computer simulation, which consisted of the 3D-structure prediction of the aptamer, molecular docking, and molecular dynamics (MD) simulation. The results revealed that MBMI-01c formed an antiparallel G-quadruplet (GQ) structure, and CTX-MI was tightly bound in the groove of the GQ structure by four hydrogen bonds. Additionally, a BLI-based aptasensor was developed and applied to CTX-MI detection in tap water. The aptasensor showed the potential for the rapid detection of CTX-MI in a concentration of 1–50 μM with good precision and reproducibility.

## 2. Results and Discussion

### 2.1. The Isolation of Aptamers for CTX-MI

MB-SELEX included positive selection and counter selection (Figure 1A), and the counter selection was an optional step to remove the nonspecific adsorption to MBs or analogue α-conotoxin GI (CTX-GI). To improve the enrichment of the dominant sequences, the screening pressure was increased by reducing the amount of the CTX-MI coated MBs (CTX-MI-MBs), reducing the incubation time, and increasing the washing time after incubation. The recovery ratio was significantly decreased at pressurized rounds (odd rounds), which proved that the sequences with weaker affinity were effectively removed (Figure 1B). Counter selection was introduced at round 7; blank MBs and CTX-GI-coated MBs (CTX-GI-MBs) were used for rounds 7–9 and 10–12, respectively. The counter selection was effective, as the counter recovery ratios greatly decreased (Figure 1B). After 11 rounds of selection, the recovery ratio reached a plateau at 1.06% and showed no significant increase at round 12 (Figure 1B); hence, the screening process was terminated.

### 2.2. The Selection of Candidate Aptamers

Generally, the proportion of high-affinity aptamers presents a prominent preponderance after screening and PCR amplification. Therefore, compared with random clone sequencing in most previous reports [20,21,22], the HTS analysis reflects the enrichment of the dominant sequences and the variation in the sequence diversity more intuitively [23], which is more conducive to acquiring prominent aptamers. The libraries of rounds 2, 4, 6, 9, and 12 were chosen for HTS analysis. Rounds 2, 4, and 6 were the enriched rounds following the pressured rounds. Round 9 was the end of the blank counter selection, and round 12 was chosen as the last round.

A total of 138,149 unique sequences were detected. The pool of round 2 was highly diverse, containing 92.1% of unique sequences. With the increase in screening pressure, the sequence diversity was significantly decreased to 36.5% in round 12, which indicated the elimination of weak-affinity sequences (Figure 2A). Next, we retrieved the sequences with a frequency greater than 0.1% in round 12 as the prominent ssDNA (a total of 98 sequences). The percentage of prominent sequences increased from 6.9% (round 2) to 60.1% (round 12) (Figure 2B), indicating that the dominant sequences were amplified and enriched gradually with each round. An 18.2% reduction in sequence diversity and a 21.5% increase in the frequency of prominent aptamers were analyzed, although the condition of the positive selection in rounds 10–12 was the same as in round 9, demonstrating that the nonspecific aptamers were effectively eliminated (Figure 2A,B). The evolution of the 10 most enriched sequences is shown in Figure 2C, and the detailed information on the top 10 sequences is shown in Appendix A. The highest-enriched sequence (MBMI-01) accounted for a frequency of 10.9% in round 12, and these 10 sequences showed a trend of enrichment, which also indicated the conditions were favorable for the enrichment of candidate ssDNA.

Subsequently, multiple sequence alignment was performed using Clustal X 2.1, and the neighbor-joining tree (N-J tree) was constructed by MEGA 7 (Appendix A). These 98 sequences were grouped into four families (I–IV) based on the N-J tree. The homologous sequences showed a wide range of conserved regions, which were probably related to the recognition of the target. Free energy was predicted using the Mfold web server (http://www.unafold.org/mfold/applications/dna-folding-form.php) (accessed on 15 September 2021); then, the aptamers with a higher frequency and lower free energy were chosen (Appendix A).

### 2.3. The Determination of Affinity and Specificity

The affinity between the candidate aptamers and marine toxins was detected by BLI assay. Firstly, 10 μM of CTX-MI was used for preliminary affinity detection. The results are shown in Appendix A. Nine sequences (marked in red) had an affinity with CTX-MI with constant (K_D_) values ranging from 0.26 to 21.2 μM. MBMI-01, MBMI-02, and MBMI-92 showed a relatively high affinity and response value in each family. The 10-bp complementary base pairs in the fixed regions could reduce the cross-linking between chains and promote the independent fold of random regions. Previous reports also showed the negligible contribution of fixed regions to the aptamer-target interaction [22]. Therefore, new aptamers (named MBMI-01c, MBMI-02c, and MBMI-92c) were resynthesized without the fixed regions; no significant change in the K_D_ values and a slight increase in response values were detected (Appendix A). MBMI-01c, which had the highest response (0.2317 nm) and lowest K_D_ value (0.87 μM), was chosen for further determination.

Thereafter, CTX-MI with a series of concentration gradients (1.25, 2.5, 5, 10, and 20 μM) was used to accurately determine the affinity of MBMI-01c; and the K_D_ value was 0.524 μM (Figure 3A). Specificity detection was conducted along with CTX-GI, domoic acid (DA), okadaic acid (OA), saxitoxin (STX), and tetrodotoxin (TTX), and the final concentration of each toxin was 10 μM. CTX-GI is the analogue of CTX-MI, which is considered as the second most toxic CTX [24]. The DA, OA, STX, and TTX are the typical detection indexes of marine toxins. As shown in Figure 3B, high response values with no significant difference were monitored when CTX-MI or the mixture of the aforesaid six toxins were added, while there was almost no response in other groups, suggesting that MBMI-01c was bound to CTX-MI with specific high affinity.

### 2.4. The Interaction Mechanism between MBMI-01c and CTX-MI

Based on computer simulation, the 3D structure of the MBMI-01c and the interaction mechanism between MBMI-01c and CTX-MI were simulated. MBMI-01c had the basic body to form GQ, which might be the core structure combined with CTX-MI. To build the 3D model of MBMI-01c, we first identified the composition bases of the G-tetrads by quadruplex forming G-rich sequences (QGRS) prediction. There were four possible QGRS sequences (named A to D) with the highest G-scores of 19 (Appendix A). After extending the non-GQ-core segments, we obtained nine candidate full-length models for each QGRS sequence named Q1 to Q5 and Q14 to Q17 (Appendix A). A temperature-dependent MD (TdMD) simulation was further carried out, and AQ1 (the Q1 model of structure A) proved to be the most stable model with the slightest root mean square deviation (RMSD) fluctuation (Appendix A). We also mutated guanines to cytosines in MBMI-01c to preserve particular GQ structures, and the mutational sequences were named M_1_ to M_3_ (Appendix A). The invariant response value and K_D_ value were only detected in M_2_, which only formed structure A with the highest G-scores (Appendix A). Therefore, structure A was the primary structure to interact with CTX-MI compared with structures B, C, and D. Additionally, reduced affinity was detected in M_3_, which perhaps related to the formation of the dominant structure E (Appendix A). Thus, AQ1 conformation at the end of the constant-temperature phase was employed as the native model (Figure 4A). MBMI-01c formed an antiparallel GQ with two G-planes (G4-G11-G26-G17 and G5-G10-G18-G25), which were connected by a π bond (Figure 4A,B and Appendix A).

Thereafter, the interaction mechanism between MBMI-01c and CTX-MI was revealed using molecular docking. The E_total_, E_shape_, and E_force_ are the main parameters for evaluating binding capacity. E_total_ and E_shape_ are inversely proportional to the hydrogen bonding energy and the spatial matching degree, respectively, while the lower E_force_ indicates the smaller mutual exclusion. As shown in Table 1, CTX-MI was evaluated as the tightest ligand with the lowest E_total_ value (−531.44 kcal/mol) and E_shape_ value (−531.44 kcal/mol) compared with other toxins. As shown in Figure 4C, CTX-MI was bound to the groove at the bottom of the GQ structure with hydrogen bonds. The T3, G4, C13, and T37 in MBMI-01c formed hydrogen bonds with the Gly1, Pro6, Cys4, and Lys10 in CTX-MI, respectively, and the H-bond distances were 2.7, 2.4, 2.7, and 1.9 Å, respectively (Figure 4C). The interaction between the G4 and Pro6 also provided a stable binding site at the top of GQ. Thus, CTX-MI was firmly adsorbed by the two G-planes and fixed in the groove. Conversely, hydrogen bonds were only observed between one of the G-planes and other toxins, even though more were formed with CTX-GI or STX, which led to lower spatial matching degrees, longer H-bond distances, and weaker interactions (Appendix A). MD simulation was further studied to evaluate the structural stability. As shown in Figure 4D, the equilibrium state was maintained after 5 ns. The MBMI-01c-CTX-MI compound was the most stable with the lowest RMSD values.

Hence, the favorable and specific combination between MBMI-01c and CTX-MI was revealed by molecular docking and MD simulation, which was also confirmed by the results of the BLI assay (Figure 1D). The GQ structures have great thermal stability and specific recognition, involved in several biological processes such as the regulation of gene expression and cancer induction [25,26,27,28]. In addition, the GQ structures are employed as biosensors, recognition elements, and nanomaterials [29,30,31]. Our results also revealed that the GQ structure of MBMI-01c could bind to CTX-MI with high specificity, indicating that MBMI-01c was an alternative recognition element for CTX-MI.

### 2.5. The Development of a BLI-Based Aptasensor for CTX-MI Detection

As a novel recognition element, MBMI-01c was further carried out for CTX-MI detection. The BLI platform is widely used for analyzing interactions because of several excellent characteristics, such as being label-free and real-time with high sensitivity [32,33]. The binding of receptors and ligands causes changes in the optical thickness and mass density, which leads to the shift in the interference spectrum on the surface of sensors and reflects as a response value and affinity constant [33]. When the sensor was saturated with the aptamer, the response value was proportional to the concentration of the target in the linear range. According to this principle, we reported the first BLI-based aptasensor [20], and this technology has been rapidly developed in recent years [34,35]. Hence, MBMI-01c was used to construct a BLI-based aptasensor. The aptasensors were super streptavidin (SSA) sensors saturated with biotin-modified MBMI-01c; the procedure for aptasensor preparation and sample detection was similar to the description in Section 4.4. Additionally, the response value should reach the plateau before the end of the association step; otherwise, the samples should be diluted. In addition, the buffer at the dissociation step should be consistent with the association step. Following the procedure, an independent detection was finished in 10 min.

Next, we assessed the properties of the aptasensor. CTX-MI in gradient concentrations of 0.5–100 μM was measured, and the response values were fitted to the equation: y = (2.849 − 0.02272)/[1 + (x/155.6)^−0.9271^] + 0.02272 with an R^2^ of 0.9928 (Figure 5A), indicating the dose dependence of the response values. In addition, the linear range was 1–50 μM, which was expressed by the equation: y = 0.0142x + 0.0792 with an R^2^ value of 0.9981 (Figure 5B). Then, the standard deviation of the blank response was calculated based on ten independent detections. The limit of detection (LOD) and limit of quantification (LOQ) were 0.26 and 0.88 nM, respectively. Moreover, the standard solutions in the concentration of 5, 10, and 20 μM were detected. The actual detection values were 5.48, 10.21, and 20.14 μM, while the coefficients of variation (CV) were 3.25%, 2.45%, and 2% (Table 2), respectively, indicating that the aptasensor possessed good precision. Additionally, the anti-interference of the aptasensor was ensured by the specificity of MBMI-01c (Figure 3B) and the consistent buffer at the association and dissociation steps. To verify the influence of the matrix effect on the trueness, tap water with CTX-MI was tested, and the results are shown in Table 3. A good recovery percentage of 98.83% to 102.11% was obtained, while the CV values were 4.31%, 4.9%, and 2.21%, respectively. In summary, the aptasensor provided accurate and rapid quantification of CTX-MI at the range of 1–50 μM with good precision, trueness, and repeatability.

At present, the methods for CTX-MI detection are extremely scarce; only high-performance liquid chromatography (HPLC) and matrix-assisted laser desorption/ionization mass spectrometry (MALDI-MS) have been reported [13,36]. The former is more widely used in the purification of synthetic CTX-MI [12,36]. The latter is widely used to identify peptides and proteins, which was proved feasible for CTX-MI detection according to Yasushi’s report [13]. Although these methods are considered as the gold standard, the complicated process of sample preparation and detection, the extensive use of chemical reagents, and the expensive instruments limit their application scenarios. Although some results have shown that the detection time can be reduced to 40 min [13], these methods are still not conducive to mass screening and rapid detection. Although the detectable concentration range is comparable to previous methods [13], the BLI-based aptasensor has the following advantages: (1) A simplified process of sample preparation: the interference of the matrix on the trueness can be effectively eliminated because of the same buffer at the association and dissociation steps and the specific interaction between MBMI-01c and CTX-MI. Therefore, the process of sample extraction, purification, and enrichment can be simplified. (2) Rapid detection: the aptasensor preparation and sample detection can be completed in 10 min. In addition, the platform with eight channels can be used for simultaneous detection. (3) Low cost: the simplicity for the operator, the low-budget synthesis of aptamers, and the reproducible SSA sensors reduce the human and experimental costs. (4) Portability: the portable BLI platform makes in situ detection possible. Therefore, this aptasensor is more suitable for rapid and mass detection.

## 3. Conclusions

CTX-MI is the most toxic CTX, which poses a great threat to underwater operations [10]. This study was the first report of DNA aptamer selection and identification for CTX-MI. MBMI-01c, the aptamer for CTX-MI, was predicted to form a stable antiparallel GQ, which provided a groove to capture CTX-MI specifically. As confirmed by previous reports, the groove structure provides a binding platform for small molecules [31,37,38]. Subsequently, a BLI-based aptasensor was proposed, which has good accuracy and reproducibility in the concentration range of 1–50 μM. The advantages of this aptasensor are the simple process for sample and sensor preparation, rapid detection with multiple channels, lower cost, good accuracy and repeatability, and anti-interference. Therefore, this aptasensor is an alternative tool to detect CTX-MI for rapid warning and poisoning diagnosis.

In recent years, a large number of aptamers for small molecules have been screened and applied in drug development [39,40], experimental research [16], and many other fields. According to the results of the MD simulation and BLI assay, we confirmed the good selectivity and affinity of MBMI-01c and the stability of the compound. Hence, as the novel recognition element, MBMI-01c can be applied for the clearance of CTX-MI, for the development of targeted antidotal agents, and as a probe for the research into toxicity mechanism and signal transmission.

More importantly, it is worth mentioning that the wider varieties of CTXs are also known as “the treasury of marine active molecules” because of their diverse biological activities [41]. Some of them have the potential for analgesia, antisepsis, anticonvulsion, and drug withdrawal, which are either currently used or have been tested in clinical trials [2,41,42,43,44]. Aptamers for these CTXs provide powerful tools for screening the medicinal species, identifying the active structures, and enriching the active components. Hence, scientific systematic and efficient methods for aptamer screening and characterizing are required. The programmed process we used, which was developed by MB-SELEX, HTS analysis, affinity detection, and computer simulation, was proved to be effective for aptamer screening and characterization for CTX-MI. The dominant sequences were effectively enriched while the low-affinity and nonspecific ssDNA were eliminated through progressively pressurized MB-SELEX. The variations in sequence diversity and frequency were intuitively characterized by HTS analysis, which could scientifically determine the dominant sequences. The programmed computer simulation, which was composed of 3D structure prediction, molecular docking, and MD simulation revealed detailed structural information and binding modes in the natural state. Associated with experimental affinity detection, the specific and efficient identification between the aptamers and targets was comprehensively described. More importantly, the contribution of each nucleotide toward recognition and affinity was visually displayed, which was helpful for the subsequent modification and improvement of the aptamers. Hence, the programmed process is applicative to other CTXs.

## 4. Materials and Methods

### 4.1. Chemicals and Reagents

CTX-GI and CTX-MI were purchased from EFEBIO (Shanghai, China). OA, STX, and TTX were purchased from Taiwan Algal Science Inc. (Taiwan, China). DA was purchased from Sigma-Aldrich Co. LLC (Shanghai, China). All ssDNA oligonucleotides were synthesized and purified by Sangon Biotechnology Co. Ltd. (Shanghai, China). Qubit^®^ ssDNA Assay Kits, Pierce^TM^ Quantitative Fluorometric Peptide Assay Kits, Dynabeads^®^ M-270 Carboxylic Acid, and the relevant reagents were purchased from Thermo Fisher Scientific Inc. (Waltham, MA, USA). GoTaqHot^®^ Start Colorless Master Mix was purchased from Promega Corporation (Madison, WI, USA). The QIAEX^®^ II Gel Extraction Kit was obtained from Qiagen (Frankfurt, Germany). SSA sensors were purchased from ForteBio (Shanghai, China). The selection Buffer (SB) (0.9 mM CaCl_2_, 2.7 mM KCl, 1.5 mM KH_2_PO_4_, 0.6 mM MgCl_2_·6H_2_O, 0.1 M NaCl and 20 mM Na_2_HPO_4_, pH = 7.4) was purchased from Tiandz (Beijing, China). The SB was used for aptamer screening, the BLI assay, and the performance evaluation of the aptasensor.

### 4.2. Aptamers Selection In Vitro

#### 4.2.1. Preparation of MBs

According to the specification, CTX-MI or CTX-GI was coupled with the MBs for positive and counter selection, respectively. The N-terminal of CTX-MI and CTX-GI has a free amino group, which could form amide bonds with carboxyl on the surface of the Dynabeads^®^ M-270 Carboxylic Acid. Briefly, CTX-GI or CTX-MI was dissolved in MES solution (25 mM, pH = 5.0) to 21 mM. Then, 50 mg of the EDC or NHS was dissolved in 1 mL MES solution. Next, 100 μL of MBs was washed twice with the MES solution and incubated with 50 μL of the EDC solution and 50 μL of the NHS solution at room temperature for 30 min to activate the superficial carboxyl groups. After removing the supernatant, 100 μL of the CTX-GI or CTX-MI solution was added and incubated for 1 h. After that, the supernatant was quantified using Pierce^TM^ Quantitative Fluorometric Peptide Assay Kits (Waltham, MA, USA) to calculate the efficiency of the immobilization. In order to quench the non-reacted activated carboxylic acid groups, the aforesaid MBs were incubated with 50 mM Tris (pH = 7.4) for 1 h. The coated MBs were collected, mixed with 100 μL SB, and stored at 4 °C. The blank MBs were also activated and quenched for counter selection. In this study, the concentrations of CTX-MI-MBs and CTX-GI-MBs were 14.7 μM and 16.2 μM, respectively.

#### 4.2.2. Random ssDNA Library and Primers

The sequences of the random ssDNA library and primers are shown in Table 4. An 80 nt random ssDNA library (Lib V_1_) was used, which consisted of a central region of 40 nt random nucleotides and 20 nt fixed nucleotides at both the 3′ and 5′ ends. The fixed nucleotides were primer binding sites for the amplification of the libraries, which were binding with F_1_ and R_1_, respectively. In addition, the fixed regions at both ends can form a complementary pairing of 10 bp. The R_1_ was designed with a poly adenine tail (A_20_ = 20 nt A), which facilitated the separation of the two different-sized PCR products. The primers of F_2_–F_6_ and R_2_ were used to obtain the HTS samples.

#### 4.2.3. Aptamer Selection by MB-SELEX

MB-SELEX was conducted according to the previous report [34]. The process is illustrated in Figure 1A, following the protocol detailed in Appendix A. The ssDNA libraries were dissolved, denatured at 95 °C for 10 min, and cooled at 4 °C for 5 min to ensure the optimal structural conformation. The prepared libraries were added into the CTX-MI-MBs, filled with SB to 100 μL, and incubated at room temperature. During incubation, the ssDNA gradually formed specific tertiary structures. The sequences with affinity were immobilized on the CTX-MI-MBs by binding to CTX-MI. Then, the beads were incubated with SB to rinse the unbound ssDNA. The ssDNA combined with CTX-MI-MBs was eluted with 200 μL SB by heating at 95 °C for 30 min and quantified for the recovery ratio. Then, the eluted ssDNA was amplified by PCR, and the amplified ssDNA was prepared and collected from the PCR products for the next round. For the counter selection, the ssDNA libraries were incubated with counter beads (blank MBs or CTX-GI-MBs) to eliminate the nonspecific binding, and then the supernatants were recovered for the following positive selection.

### 4.3. The Preparation of HTS Samples

After 12 rounds of selection, the enriched libraries of rounds 2, 4, 6, 9, and 12 were amplified with different forward primers (F_2_–F_6_) and the same reverse primer (R_2_) to carry different tags (Table 4). The PCR products in each pool were purified and mixed in equal quantities for the HTS analysis.

### 4.4. The Determination of Affinity by BLI Assay

The BLI assay was conducted using the OctetRED K2 instrument (Sartorius, Shanghai, China) to determine the affinity parameter and response value. The procedure followed the previous report [34], and the details are shown in Appendix A. The procedure included five steps: baseline (1 min), loading (3 min), baseline-2 (1 min), association (2 min), and dissociation (3 min). Baseline and baselin-2 were balancing steps immersing with SB. The biotin-modified aptamers (2 μM) were immobilized on the SSA sensors at the loading step. After the baseline-2 step, the sensors were subjected to the association step, immersing with targets diluted in SB. Following the association step, the sensors were immersed in SB to estimate the dissociation. All the steps were performed at 25 °C. Toxin solutions were replaced with SB to exclude the effect caused by the buffer and sensors. The ForteBio Data Analysis 11.0 Software (Sartorius, Shanghai, China) was used to analyze the data. The K_D_, association constant (K_on_), and dissociation constant (K_dis_) were estimated to establish the binding affinity between the toxins and aptamers. The K_D_ value was determined utilizing the 1:1 global fit model. According to the instrument instructions, the result is considered valid only if the K_D_ error is at least an order of magnitude less than the K_D_ value, X^2^ < 3, and R^2^ > 0.8. Otherwise, the result is a false-positive and is considered a non-affinity.

### 4.5. Computer Simulation

#### 4.5.1. Three-Dimensional Structure Prediction of MBMI-01c

Firstly, the sequence of MBMI-01c was submitted to the QGRS Mapper web server (http://bioinformatics.ramapo.edu/QGRS/analyze.php) (accessed on 28 June 2022) to predict the composition and distribution of the QGRS [45]. The G-score was output according to the possibility of forming stable GQ. We further predicted all possible 3D structures with the highest Gscore by the G-Quadruplex module of the 3D-Nus web server (https://www.iith.ac.in/3dnus/Quadruplex.html) (accessed on 1 July 2022) [46]. Based on the output GQ-core models, the full-length 3D structures were constructed by extending and linking the residual nucleotides outside the core fragment using Discovery studio (Ver.4.6). For the conformational stability of the core fragment, Mg^2+^ ions were placed in the center of each GQ-core, and then energy minimization was carried out to remove possible geometric deficiencies using the CHARMM force fields of Discovery studio [47]. Subsequently, the thermostable model, which was treated as the natural conformation, was obtained by TdMD simulation [31].

#### 4.5.2. Molecular Docking

The structures of CTX-MI, CTX-GI, STX, and OA (.sdf) were downloaded from the NCBI Pubchem Compound database (https://www.ncbi.nlm.nih.gov/pccompound) (accessed on 9 July 2022). The structure of MBMI-01c was optimized by cleaning geometry using Discovery studio and saved as a PDB file. Then, semi-flexible docking was performed using Hex (Ver.8.0). For each docking system, the receptor and ligand were permitted to rotate 180°, and the size of the docking box was set to 10 Å.

#### 4.5.3. MD Simulation

MD simulation was performed by GROMACS with the SPC water model. In each system, the model of the aptamer was placed in the center of a water box (20 Å^3^); the solvent layer between the edge of the box and the solute surface was set to 10 Å; Mg^2+^ and Cl^−^ ions were randomly placed by replacing water molecules to neutralize the system. The parameter file was configured according to the AMBER99SB-ILDN force field [48]. The LINCS algorithm was employed to constrain all covalent bonds for hydrogen, and a time-step of 1.0 fs was used. The remote electrostatic interaction was processed using the Particle-Mesh-Ewald (PME) method [49], and the cut-off distance of the electrostatic and van der Waals force was set to 4 Å. When dealing with the solvated system, the minimization of the steepest descent and the minimization of the conjugate gradient were 2000 cycles; all atoms were set to 50 kcal/(mol·Å^2^); the solvent molecules and hydrogen atoms were set to move freely. In the warming phase, the Langevin dynamic simulation algorithm was used to heat the system from 0 K to 300 K in 50 ps and to balance with 1 atm fixed pressure for 400 ps [50]. In the heating phase, the weak constraint of 10 kcal/(mol·Å^2^) was applied to all atoms, and the Velocity rescaling [51] algorithm was used to group the aptamers, toxins, and aptamer-toxin complexes. Each periodic boundary dynamics simulation was performed at a controlled temperature of 300 K and a controlled pressure of 1 atm. The variation curves of the RMSD values were derived after a 100 ns simulation to judge the stability of the conformation.

### 4.6. Performance Evaluation of the BLI Aptasensor

To characterize the practicability of this BLI aptasensor, the linear range, LOD, LOQ, precision, specificity, accuracy, and feasibility were tested according to the previous reports [20,52]. The linear range was measured by testing the CTX-MI standards in the concentration range of 0.5–100 μM. The relation between the response value and the CTX-MI concentration was described by the sigmoidal logistic four-parameter equation, as follows:y = (R_max_ − R_min_)/[1 + (x/EC_50_)^b^] + R_min_(1)
where R_max_ and R_min_ are the maximum and minimum response values, respectively. EC_50_ represents the concentration at which the response value reaches half of the maximum response value, and b is the slope of the curve. The linear range was obtained according to the aforesaid curve and was considered satisfactory if R^2^ > 0.99. The LOD and LOQ were determined by testing the blank groups (*n* = 10) according to the following equations:LOD = 3S/b(2)
LOQ = 10S/b(3)
where S is the standard deviation of the response value, and b is the slope of the linear regression equation. The precision was estimated with the CV by testing the CTX-MI standards in the concentrations of 5, 10, and 20 μM (*n* = 3). The accuracy and feasibility were estimated by the standard recovery rate in tap water, and the concentrations of CTX-MI were 10, 20, and 25 μM (*n* = 3).

### 4.7. Treatment of Real Samples

Tap water samples with different concentrations of CTX-MI standards were prepared according to Ouyang’s report [34]. Briefly, 1 mL of tap water spiked with CTX-MI standards was evaporated, resuspended with 1 mL SB, and filtered through the 0.45 μm filters.

### 4.8. Statistical Analysis

Statistical analysis was performed by Microsoft Excel 2010 or GraphPad Prism 6.0. The data were subjected to analysis by Student’s *t*-test using SPSS version 22.0. The mean differences were considered significant at *p* < 0.05.

## Figures and Tables

**Figure 1 toxins-14-00706-f001:**
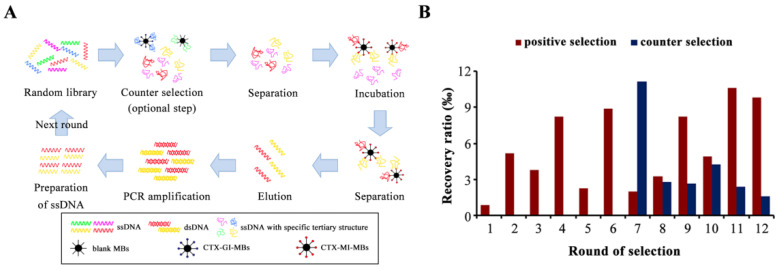
Aptamer selection in vitro. (**A**) The workflow of the magnetic beads SELEX (MB-SELEX). (**B**) The recovery ratio of ssDNA in the MB-SELEX. The recovery rate represents the ratio of the amount of ssDNA eluted from the magnetic beads to the amount of the ssDNA input in this round.

**Figure 2 toxins-14-00706-f002:**
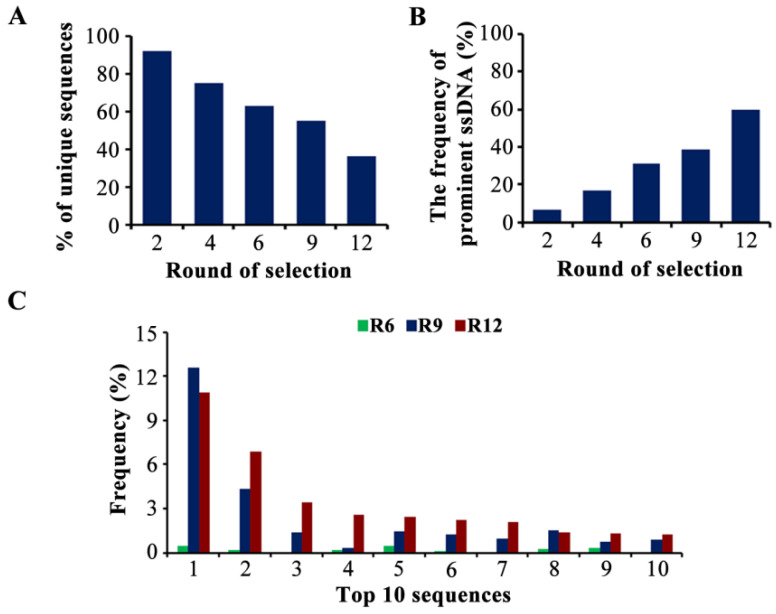
The analysis of the high-throughput sequencing (HTS) data. (**A**) The sequence diversity in each pool. The ordinate represents the number of unique sequences in each pool as a percentage of the total number of unique sequences. (**B**) The frequency of the prominent ssDNA in each pool. The ordinate represents the reads of the prominent ssDNA as a percentage of the total reads in each pool. (**C**) The frequency of the top 10 sequences. The ordinate represents the reads of each sequence as a percentage of the total reads in each pool.

**Figure 3 toxins-14-00706-f003:**
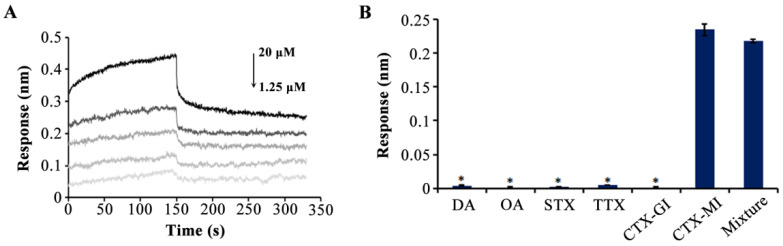
The affinity and specificity of MBMI-01c. (**A**) Association and dissociation curves after the addition of α-conotoxin MI (CTX-MI) (the concentrations of CTX-MI corresponding to the five curves were 1.25, 2.5, 5, 10, and 20 μM from bottom to top). The association step and dissociation step were 0–150 s and 150–300 s, respectively. (**B**) The response value for each target (10 μM). The target in the mixture group was the mixture of these six toxins (10 μM). Error bar represents the standard deviations. * *p* < 0.05 vs. CTX-MI group. All response values were detected by the BLI assay.

**Figure 4 toxins-14-00706-f004:**
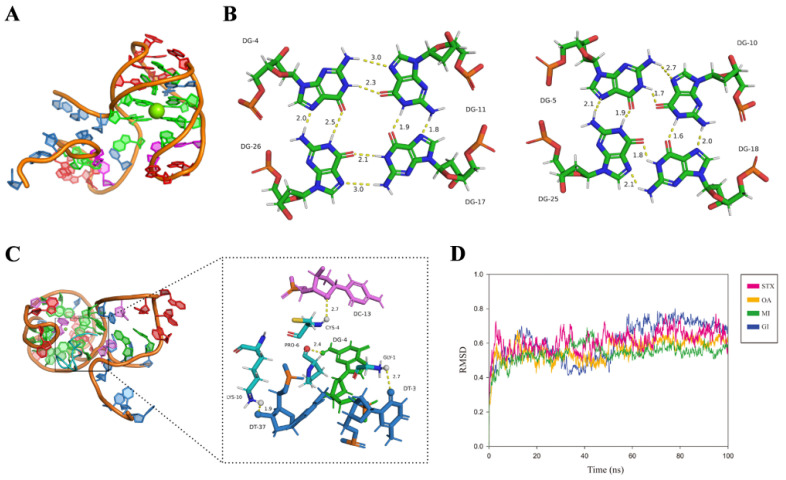
The results of the computer simulation. (**A**) The full-length 3D model of MBMI-01c. (**B**) The three-dimensional structure of two quadruplets of the MBMI-01c model. (**C**) The three-dimensional view of the interaction between the MBMI-01c and CTX-MI. The hydrogen bond between the docking site of the MBMI-01c and CTX-MI is depicted by a yellow dashed line, and the distance was marked next to the dashed line. (**D**) The root mean square deviation (RMSD) of the MBMI-01c-toxin compound vs. the simulation time. The names of toxins are noted in the legend.

**Figure 5 toxins-14-00706-f005:**
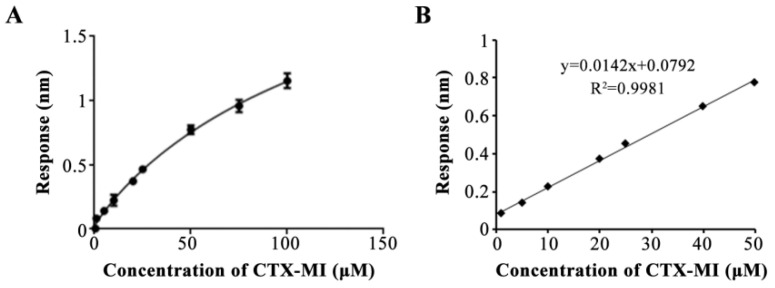
Curve fitting for the response value and CTX-MI concentration. (**A**) The calibration curve for CTX-MI from 0.5 to 100 μM, a plot of the response value as a function of the CTX-MI concentration. The error bars represent standard deviations. (**B**) The linear range of the calibration curve for CTX-MI, a plot of the response as a function of the CTX-MI concentration from 1 to 50 μM. All response values were detected by the BLI assay.

**Table 1 toxins-14-00706-t001:** The results of semi-flexible docking when MBMI-01c was set as the target.

Ligand	E_total_ (kcal/mol)	E_shape_ (kcal/mol)	E_force_ (kcal/mol)	RMS
CTX-MI	−531.44	−531.44	0	−1.00
CTX-GI	−439.61	−439.61	0	−1.00
OA	−264.49	−264.49	0	−1.00
STX	−388.01	−388.01	0	−1.00

**Table 2 toxins-14-00706-t002:** Precision studies with different concentrations of CTX-MI (*n* = 3).

CTX-MI (μM)	Detection Value (μM)	CV (%)
5	5.48	3.25
10	10.21	2.45
20	20.14	2.00

**Table 3 toxins-14-00706-t003:** Recovery studies of tap water with different concentrations of CTX-MI (*n* = 3).

CTX-MI (μM)	Detection Value (μM)	CV (%)	Recovery (%)
10	10.47	4.31	102.11
25	24.71	4.90	98.83
50	50.53	2.21	101.06

**Table 4 toxins-14-00706-t004:** The sequence of the random sDNA library and primers.

ID	Sequence (5′ to 3′)
Lib V_1_	ATTGGCACTCCACGCATAGG-N_40_-CCTATGCGTGCTACCGTGAA
F_1_	ATTGGCACTCCACGCATAGG
F_2_	AAAGCAATTGGCACTCCACGCATAGG
F_3_	AACGCCATTGGCACTCCACGCATAGG
F_4_	AAGGCGATTGGCACTCCACGCATAGG
F_5_	ACAGGAATTGGCACTCCACGCATAGG
F_6_	ACCGGCATTGGCACTCCACGCATAGG
R_1_	A_20_-Spacer_18_-TTCACGGTAGCACGCATAGG
R_2_	TTCACGGTAGCACGCATAGG

## Data Availability

Not applicable.

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
