# Peer review of "Programmed Aptamer Screening, Characterization, and Rapid Detection for α-Conotoxin MI"

_toxins, 2022, doi:10.3390/toxins14100706_

Round 1
Reviewer 1 Report
The present work is an interesting and well-made research piece, with clear applicative potentialities. However, it is incorrectly/badly or strangely written in many parts of it, from the abstract to the conclusions, a fact which generates loss of attraction and difficult understanding for readers and must mended. A general revision of the English language would be required.
Among others, examples of such incorrect or inadequate uses are in lines 6 (... lack of drugs ... probably meaning "lack of counteracting drugs"), 8 (deadlock), 9 (hence/here?/in this work?), 11 (performed), 20 (was/were), 22 (organized?), 26 (gastropod/gastropods), 29 (cysteine/ cysteines), 30 (the species/the variants/the variant species ...), 33 (minced/split/branched ...), 54 ("the", to be removed), 65 (to apply/to properly apply?) ... etc ... 152 (determinate?), etc..., 276 (Nanomolar/Nanomolar concentrations?), 277 (underwater operations/ meaning?), 291-293 (badly constructed sentence), 302 (by?), 307 (which could scientifically determine/ is the word "scientific" required here?) ...etc ... etc.
Authors should also try to make the text and technology understandable for non-specialists readers. This is reflected in the case of the MB-Selex technology which is neither minimally o graphically well explained in the text: i.e. in Figure 1A, dealing with the schematic illustration of MB-Selex, it is not clear whether the ssDNA binds to the beads covalently by an end, which in fact is the case for the peptide/protein target/bait for which aptamers are developed/screened. It should be made clear that the peptide/protein target is the one to be covalently bound to the microbeads, and that this is achieved by the action of carbodiimide, EDC. Also, to remind that such link takes place through a given class of residues of the target (Lys? Cys?), and that EDC and NHS are reagents for such purpose. Besides, it would be worth to clarify the way to trim the aptamers, which now is poorly described (...Subsequently, the shorter aptamer was obtained by removing the fixed regions, lines 75-76. Is this enough for a proper understanding and repetition of this action, if wished?)
By the way, authors should consider the possibility of including an abbreviations section in the manuscript, given the numerous abbreviations used in it (CTX-MI, CTX-GI; OA; STX, TTX, MBMI-01, CV, LOD, LOQ ...etc).
The paper is successful in the achievement of its main goals, like getting aptamers of CTX-MI with sufficient specificity and affinity, i.e. adequate for a biosensor with many desirable properties and of easy use. However, it would also be positive to include in the Discussion a comparison with other similar developments on which the affinity of the derived aptamers reached nano- o subnanomolar levels. This could facilitate further improvements or applications (i.e. to other CTXs).
Reviewer 2 Report
The article is devoted to the study of aptamers to one of the most dangerous conotoxins - CTX-MI. The results obtained for the first time are very interesting and can be used for the development of methods for the detection of conotoxins and their neutralization. The article is well-written and detailed, the materials presented clearly illustrate all the work done. There are no serious comments.
The shortcomings (that can be easily corrected) of the work include the following
1. Figure captions are not complete and require reference to the text to understand them.
- Figure 1C and D should probably be separated into a separate one and moved to the appropriate section, the caption to it should also be deciphered in more detail, including indicating the method of measurement. In Figure 1C, indicate the time of peptide supply, and also indicate the concentration for each trace
- Fig. 4 - indicate the method of measurement
In addition, sequences from among the top 10 were not noted anywhere (Fig. 2C), perhaps they can be separately identified in the supplement materials.
Section 2.3 raised several questions
It is worth explaining in more detail what method was used to determine the affinity (it is possible to transfer some of the material about the BLI assay from Section 2.5), indicate which sensors were used specifically for the primary affinity screening.
Line 141 - 7 selected sequences are indicated, while only 6 are marked in table S4
Line 142 - there is a discrepancy between the Kd data and those indicated in table S4
From each group, the authors chose one candidate with the highest affinity, but in terms of Kd, candidate MBMI-39 is superior to MBMI-1 (group IV), perhaps it is worth explaining your choice.
Reviewer 3 Report
The manuscript describes the use of SELEX technology to develop an aptamer for binding the conotoxin CTX-MI. This ability of this aptamer to identify the conotoxin in a bio-layer interferometry platform was demonstrated. The work clearly demonstrates the promise of SELEX technology to develop molecules capable of recognising other molecules. I must say that the need for recognising conotoxins is not that great, as the authors report 300 poisoning incidents in the last 60 years, and in most incidents the cause of poisoning is likely to be known. Despite the absence of a great need for this technology the manuscript shows its capability.
In the manuscript the science needs to be better described. Figure captions need to describe what is being shown. Some of the information in the SI, such as details of the ssDNA sequences and selection buffer composition, should be included in the main manuscript.
I do not understand how the ssDNA library was enriched for sequences that bind the target CTX after each round of selection. Are the sequences that are eluted from the magnetic beads after separation treated as the next library to be screened? Or are the recovered sequences added to another collection of random sequences? If the latter case, how much is added relative to the other random sequences? Is this controlled in any way? Are the recovered sequences already present in the library that they are added to? Better explanation of this step would benefit the reader.
The English needs improvement. I list many suggestions below, but this is not complete and I recommend review by a scientific editor.
line 30 replace "the species of CTXs have" with "the number of different sequences of CTXs has"
line 33 replace "minced" with "divided"
line 37 replace "Stung by cone snails will" with "The sting of cone snails can"
line 43 replace "clinic" with "clinical"
line 94 replace "round" with "rounds"
line 96 Is the recovery ratio 1.06% or 10.6%? It looks like 10.6% on figure 1B. Is the vertical scale in figure 1B wrong?
line 98 What do the labels on the horizontal axis of figure 1B mean?
line 100 What is panel 1C showing? What sort of response is this?
line 105 I think you may mean "affinity" where "affiliative" is being used.
line 141 replace "were showed" with "are shown"
lines 108 and 115 replace "sequences diversity" with "sequence diversity"
What was the length of the sequences in the libraries? How many of these residues were kept constant at each end, and what was the length of the randomised section?
Figure S2 Do the sequences shown here include the constant regions at each end for the primers? More explanation in the figure caption is required.
line 141 Only six sequences are marked in red in table S4 and none of them has an affinity of 21.2uM.
line 146 replace "contribute" with "contribution"
Table S4 What do the columns labeled "X2" and R2" represent?
line 152 replace "determinate" with "determine"
line 156 replace "showed" with "shown"
line 158 Was CTX-MI included in the mixed group of toxins? Please clarify in text.
line 161 I don't believe any AI methods were used. It appears to be sequence analysis and molecular dynamics simulations.
line 163 replace "building" with "build"
Line 160-179 What is AQ1? What is M2? This section needs to better explain what all the constructs are, how they relate to each other, and the rationale for deriving their sequences.
line 189 What does "less mutual exclusion" mean?
line 191 replace "comparing" with "compared"
line 191 replace "showed" with "shown"
line 211 What is "HEX"?
line 221 replace "rapid" with "rapidly"
line 222 replace "saturating" with "saturated"
line 224 replace "with" with "to"
line 247 replace "detected" with "tested"
line 247 replace "were showed at" with "are shown in"
line 254 replace "scare" with "scarce"
line 262 replaced "showed" with "showing"
line 279 The three dimensional structure of the aptamer has not been "verified", only predicted. Verification would require experimental structure determination
line 321 What was the composition of the selection buffer?
line 336 replace "were showed" with "are shown"
line 347-349 I do not understand this sentence.
line 360 Why were Mg2+ ions added? Were they present in the experimental conditions?
line 371 What is the unit for the docking box, angstrom, nm?
line 375 Did the simulated Mg and Cl concentrations match the experimental concentrations? If not, why not?
line 379 replace "time" with "time-step"
line 379 delete "Ewald"
line 383 What does "all atoms were set to 50 kcal/(mol.A2)" mean? Do you mean positional restraints on all heavy atoms?
line 388 replace "compounds" with "complexes"
line 393 replace "perform" with "characterize"
line 397 replace "which as follow:" with "as follows:"
Round 2
Reviewer 3 Report
The revised manuscript incorporates many of the suggestions, making it clearer and easier to read. Most of the scientific problems have been addressed. English corrections would still benefit this version, however.
